# On the Representation Collapse of Sparse Mixture of Experts

**Zewen Chi**[†,*] **Li Dong**[‡], **Shaohan Huang**[‡], **Damai Dai**[||], **Shuming Ma**[‡], **Barun Patra**[‡]
**Saksham Singhal**[‡], **Payal Bajaj**[‡], **Xia Song**[‡], **Xian-Ling Mao**[†], **Heyan Huang**[†], **Furu Wei**[‡]
[†] Beijing Institute of Technology
[‡] Microsoft Corporation
[||] Peking University
https://aka.ms/msragi

## Abstract

Sparse mixture of experts provides larger model capacity while requiring a constant computational overhead. It employs the routing mechanism to distribute input tokens to the best-matched experts according to their hidden representations. However, learning such a routing mechanism encourages token clustering around expert centroids, implying a trend toward representation collapse. In this work, we propose to estimate the routing scores between tokens and experts on a low-dimensional hypersphere. We conduct extensive experiments on cross-lingual language model pre-training and fine-tuning on downstream tasks. Experimental results across seven multilingual benchmarks show that our method achieves consistent gains. We also present a comprehensive analysis on the representation and routing behaviors of our models. Our method alleviates the representation collapse issue and achieves more consistent routing than the baseline mixture-of-experts methods.

## 1 Introduction

Scaling up model capacities has shown to be a promising way to achieve better performance on a wide range of problems such as language model pre-training (Radford et al., 2019; Raffel et al., 2020), and visual representation learning (Dosovitskiy et al., 2021; Bao et al., 2022). Despite the effectiveness, increasing the number of parameters leads to larger computational cost, which motivates recent studies to explore Sparse Mixture-of-Experts (SMoE) models (Shazeer et al., 2017; Fedus et al., 2021; Lepikhin et al., 2021). SMoE increases the model capacity by building several sparsely-activated neural networks. With nearly constant computational overhead, SMoE models achieve better performance than dense models on various tasks, including machine translation (Lepikhin et al., 2021), image classification (Riquelme et al., 2021), and speech recognition (Kumatani et al., 2021).

The routing mechanism plays an important role in SMoE models. Given an input token, the router measures the similarity scores between each token and experts. Then we distribute tokens to the best-matched experts according to the routing scores. Recent studies explored various token assignment algorithms to improve SMoE training. For instance, Lewis et al. (2021) formulate SMoE routing as a linear assignment problem that globally maximizes token-expert similarities. Zhou et al. (2022) have experts selecting top tokens rather than assigning tokens to top experts. Roller et al. (2021) and Dai et al. (2022) propose to keep routing choices consistent. Many studies in recent years focus on how to design the token-expert assignment algorithm. In this paper, we present that current routing mechanisms tend to push hidden representations clustering around expert centroids, implying a trend toward representation collapse, which in turn harms model performance.

---

[*]Zewen Chi and Damai Dai contribute during internship at Microsoft Research.

36th Conference on Neural Information Processing Systems (NeurIPS 2022).

In order to alleviate the representation collapse issue, we introduce a simple yet effective routing algorithm for sparse mixture-of-experts models. More specifically, rather than directly using the hidden vectors for routing, we project the hidden vectors into a lower-dimensional space. Then, we apply $L_2$ normalization to both token representations and expert embeddings, i.e., measuring routing scores on a low-dimensional hypersphere. Besides, we propose a soft expert gate with learnable temperature, which learns to control the activation of experts.

We evaluate the proposed method on cross-lingual language model pre-training and fine-tuning on downstream tasks. Experimental results show that our model consistently outperforms the baseline SMoE models in terms of both language modeling and fine-tuning performance. Moreover, analysis indicates that our method alleviates the representation collapse issue compared with the SMoE baseline. Our method also achieves more consistent routing behaviors during both pre-training and fine-tuning, which confirms the effectiveness of the proposed routing algorithm.

Our contributions are summarized as follows:

- We point out the representation collapse issue in sparse mixture-of-experts models, which is under-explored in previous work.
- We propose to estimate routing scores between tokens and experts on a low-dimensional hypersphere in order to alleviate representation collapse.
- We conduct extensive experiments on cross-lingual language model pre-training and fine-tuning on downstream tasks.
- We present a detailed analysis of routing behaviors and representation properties, which shows that our method improves performance and achieves more consistent routing.

## 2 Background

### 2.1 Sparse Mixture of Experts

Sparse Mixture-of-Experts (SMoE) models take advantage of conditional computation, and have shown to be a promising way to scale up the number of parameters. In this work, we consider SMoE for Transformers, where SMoE layers are inserted into neighboring Transformer blocks. Each SMoE layer consists of a router and several expert networks. Following most previous work (Fedus et al., 2021), we use feed-forward networks as experts, instead of self-attention modules.

For the input token $x$ with its hidden representation $\boldsymbol{h} \in \mathbb{R}^d$, the router computes the routing score between $\boldsymbol{h}$ and the $i$-th expert by a dot-product similarity metric $s_i = \boldsymbol{h} \cdot \boldsymbol{e}_i$, where $\boldsymbol{e}_i \in \mathbb{R}^d$ is a learnable expert embedding, and $d$ is the hidden size of the model. Then, the router utilizes a sparse gating function $g(r)$ to make the expert network conditionally activated.

In this paper, we mainly focus on top-1 routing, i.e., only the expert with the largest routing score is activated. Formally, considering a SMoE layer with $N$ experts, the forward function of SMoE can be written as:

$$k = \arg\max_i s_i = \arg\max_i \boldsymbol{h} \cdot \boldsymbol{e}_i \tag{1}$$

$$f^{\text{SMoE}}(\boldsymbol{h}) = \boldsymbol{h} + g(s_k) f_k^{\text{FFN}}(\boldsymbol{h}) \tag{2}$$

where $f_k^{\text{FFN}}(\cdot)$ stands for the $k$-th expert network that is implemented as stacked feed-forward networks. Moreover, we explore both softmax gating (Lepikhin et al., 2021; Fedus et al., 2021) and sigmoid gating (Lewis et al., 2021; Dai et al., 2022) for the function $g(s_k)$:

$$g(s_k) = \begin{cases} \exp(s_k) / \sum_{j=1}^N \exp(s_j), & \textit{softmax gating} \\ \sigma(s_k), & \textit{sigmoid gating} \end{cases}, \tag{3}$$

where $\sigma(\cdot)$ is the sigmoid function.

### 2.2 Representation Collapse of Sparse Mixture-of-Experts

We present how representation collapse happens in sparse mixture-of-experts models. For convenience, we use $\boldsymbol{h}' = f^{\text{SMoE}}(\boldsymbol{h})$ to denote the output of the SMoE layer as in Equation (2), $S_k = g(s_k)$

to denote the $k$-th output of the softmax function, and $\boldsymbol{h}^{\text{FFN}} = f_k^{\text{FFN}}(\boldsymbol{h})$ to denote the output of the $k$-th expert network. The Jacobian matrix with respect to $\boldsymbol{h}$ is given by:

$$\boldsymbol{J} = \boldsymbol{J}_1 + \boldsymbol{J}_2 = (\boldsymbol{I} + S_k \boldsymbol{J}^{\text{FFN}}) + \sum_{j=1}^{N} S_k(\delta_{kj} - S_j)\boldsymbol{h}^{\text{FFN}}\boldsymbol{e}_j^{\top}, \tag{4}$$

where $\delta_{kj}$ is a Kronecker delta. The equation means that the Jacobian matrix can be decomposed into two terms. The first term $\boldsymbol{J}_1$ represents producing a better token representation given the current activation $S_k$. The second term $\boldsymbol{J}_2$ means to learn better gating function for appropriate activation score $S_k$. After back-propagation, the gradient is received from the above two paths, written as $\nabla_{\boldsymbol{h}}\mathcal{L} = \boldsymbol{J}_1^{\top}\nabla_{\boldsymbol{h}'}\mathcal{L} + \boldsymbol{J}_2^{\top}\nabla_{\boldsymbol{h}'}\mathcal{L}$. The second term can be expanded as:

$$\boldsymbol{J}_2^{\top}\nabla_{\boldsymbol{h}'}\mathcal{L} = \sum_{j=1}^{N} S_k(\delta_{kj} - S_j)(\boldsymbol{h}^{\text{FFN}\top}\nabla_{\boldsymbol{h}'}\mathcal{L})\boldsymbol{e}_j = \sum_{j=1}^{N} c_j \boldsymbol{e}_j, \tag{5}$$

where $c_j = S_k(\delta_{kj} - S_j)(\boldsymbol{h}^{\text{FFN}\top}\nabla_{\boldsymbol{h}'}\mathcal{L})$. The above equation indicates that *the token representation $\boldsymbol{h}$ tends to be updated toward a linear combination of the expert embeddings.*

The finding also holds for top-$K$ routing (Lepikhin et al., 2021) where the top $K$ experts ($K \leq N$) are activated for each token. The forward function of top-$K$ routing is $k_1, k_2, ..., k_K = \text{top}K(s_k)$ and $\boldsymbol{h}' = f^{\text{SMoE}}(\boldsymbol{h}) = \boldsymbol{h} + \sum_{i=1...K} g(s_{k_i})f_{k_i}^{\text{FFN}}(\boldsymbol{h})$. The gating function is defined as $g(s_{k_i}) = \exp(s_{k_i})/\sum_{j=1...K}\exp(s_{k_j})$. Similar to Equation (5), we have

$$\boldsymbol{J}_2^{\top}\nabla_{\boldsymbol{h}'}\mathcal{L} = \sum_{i=1}^{K}\sum_{j=1}^{K} S_{k_i}(\delta_{k_i k_j} - S_{k_j})(\boldsymbol{h}^{\text{FFN}_{k_i}\top}\nabla_{\boldsymbol{h}'}\mathcal{L})\boldsymbol{e}_{k_j} = \sum_{j=1}^{K} c_j \boldsymbol{e}_{k_j}. \tag{6}$$

Therefore, the above finding holds for top-$K$ routing.

We consider that such behavior potentially harms the representation capacity of Transformers. Firstly, consider that the $N$ expert vectors can span a $N$-dimensional space at most via linear combinations. As $N$ is much smaller than the hidden size $d$ in practice, the spanning subspace does not fully utilize the entire available capacity. Thus, the mechanism renders the Transformer hidden vector $\boldsymbol{h}$ collapsed to an $N$-dimensional subspace, implying a trend toward representation collapse from $\mathbb{R}^d$ to $\mathbb{R}^N$ where $N \ll d$ in practice. Secondly, Equation (5) indicates that the hidden vector $\boldsymbol{h}$ tends to be similar to the expert embedding that it is routed to. If the hidden states were routed to the same expert, they are going to be pushed closer. However, we would like to encourage the representations more diverse, so that they can be more expressive and discriminative. The phenomenon possibly restricts the expressibility of hidden states, especially when an expert is inclined to dominate routing.

## 3 Methods

We introduce the routing algorithm for sparse mixture of experts, which measures the routing scores between tokens and experts on a low-dimensional hypersphere. As shown in Figure 1b, we address the representation collapse issue of SMoE by applying dimensionality reduction and $L_2$ normalization for the token representations and expert embeddings. Then, we describe how to incorporate the routing algorithm into an SMoE model under the pre-training-then-fine-tuning paradigm.

### 3.1 Routing Algorithm

**Dimension Reduction** In order to alleviate the representation collapse issue mentioned in Section 2.2, we represent the expert embedding $\boldsymbol{e}_i$ and the token vector $\boldsymbol{h}$ in a low-dimensional space instead of the original high-dimensional hidden space. Specifically, we first parameterize the experts with lower-dimensional embeddings $\boldsymbol{e}_i \in \mathbb{R}^{d_e}$ such that $d_e$ is much smaller than the Transformer hidden size $d$. Next, we conduct a projection over the hidden states $f^{\text{proj}}(\boldsymbol{h})$, which projects $\boldsymbol{h}$ to the expert embedding space. We use a linear projection $f^{\text{proj}}(\boldsymbol{h}) = \boldsymbol{W}\boldsymbol{h}$ such that $\boldsymbol{W} \in \mathbb{R}^{d_e \times d}$. Thus, the routing scoring function between the tokens and experts can be written as $s_i = (\boldsymbol{W}\boldsymbol{h}) \cdot \boldsymbol{e}_i$. Typically we set $d_e = N/2$ (i.e., half of the number of experts) in our implementation.

Inspired by Jing et al. (2022), dimension reduction mitigates the issues described in Section 2.2 from two perspectives. First, linear projection $\boldsymbol{W}\boldsymbol{h}$ isolates the direct interaction between hidden vector $\boldsymbol{h}$

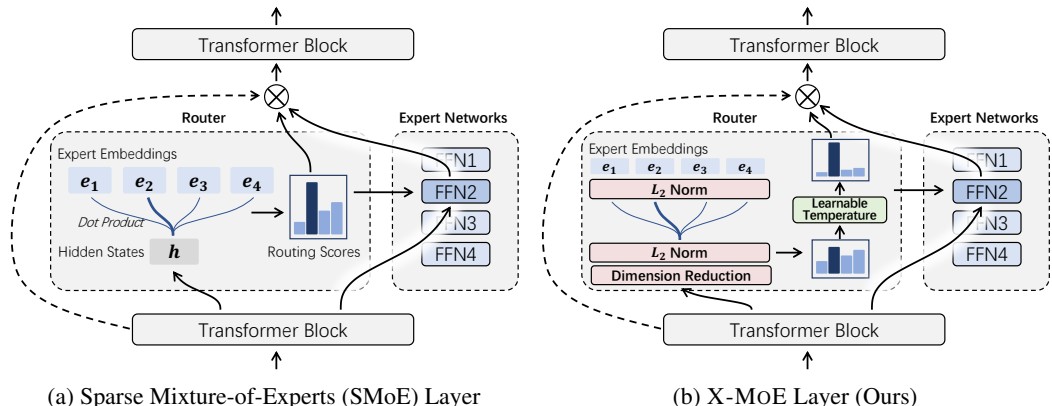

(a) Sparse Mixture-of-Experts (SMoE) Layer          (b) X-MOE Layer (Ours)

Figure 1: Illustration of a typical SMoE layer and the proposed X-MOE layer. (a) An SMoE layer consists of a router and expert networks, where the experts are sparsely activated according to dot-product token-expert routing scores. (b) X-MOE improves the routing algorithm via dimension reduction, $L_2$ normalization, and gating temperature.

and expert embedding $e_i$, which tends to relieve cascaded collapse for representations. Second, it is natural to apply a low-rank projector for hidden vectors, as the number of experts is usually much smaller than the hidden size of Transformers. Hence the reduced dimension better fits in with the low-rank nature of routing.

$L_2$ **Normalization**    After dimension reduction, we apply $L_2$ normalization to both token representations and expert embeddings. Our routing score is defined as:

$$s_i = \frac{(\boldsymbol{W}\boldsymbol{h}) \cdot \boldsymbol{e}_i}{\|\boldsymbol{W}\boldsymbol{h}\|\|\boldsymbol{e}_i\|}, \tag{7}$$

where $\|\cdot\|$ is $L_2$ normalization. Thus, the resulting representations are transformed into a certain scale with stabilized routing scoring.

As described in Section 2.2, if an expert dominated a set of hidden states, the representations were pushed toward the expert embedding. In order to fully utilize the space, we favor larger uniformity of representations while avoiding dominated experts. Given a hidden vector $\boldsymbol{h}$, the dot-product routing $s_i = (\boldsymbol{W}\boldsymbol{h}) \cdot \boldsymbol{e}_i$ is affected by both $\|\boldsymbol{e}_i\|$ and $\cos(\boldsymbol{W}\boldsymbol{h}, \boldsymbol{e}_i)$. So some experts are allocated with more tokens because of larger values of $\|\boldsymbol{e}_i\|$. In contrast, $L_2$ normalization projects vectors on the unit hypersphere, which suppresses the undesired effect of $\|\boldsymbol{e}_i\|$. The visualization in Figure 2b also confirms that our method improves the uniformity of learned representations.

*Implementation Tips.* When scaling the model to more experts, empirically, we observe that the resulting token assignments can be in a fluctuation if the expert embedding norm $\|\boldsymbol{e}_i\|$ is small. Therefore, we initialize the expert embeddings with $L_2$ norm of $0.1$ and keep the norm unchanged during training. Since the expert embeddings are parameterized in the space of $\mathbb{R}^{d_e}$, the change rate of the angle of $\boldsymbol{e}_i$ is in inverse proportion to $\|\boldsymbol{e}_i\|$. As a result, if the norm is small, the angle of $\boldsymbol{e}_i$ is updated fast, finally leading to the fluctuation of token assignments, especially when scaling up the model with more experts.

**Gating with Learnable Temperature**    In addition, we add a learnable temperature scalar $\tau$ in the SMoE gating function $g(s_k)$. Because $L_2$ normalization rescales the routing scores $s_k$ to the range $[-1, 1]$, directly using the scores for SMoE gating tends to make expert activation too conservative. The introduced temperature enables the router to adjust the gating $g(s_k)$ accordingly. To be more specific, our gating function is:

$$g(s_k) = \begin{cases} \frac{\exp(s_k/\tau)}{\sum_{j=1}^{N} \exp(s_j/\tau)}, & \textit{softmax gating} \\ \sigma(s_k/\tau), & \textit{sigmoid gating} \end{cases}, \tag{8}$$

where $\sigma(\cdot)$ is the sigmoid function, and the temperature scalar $\tau$ is learnable.

## 3.2 Training Objective

The training objective is jointly minimizing the loss of the target task and an auxiliary load balancing loss (Fedus et al., 2021). The load balancing loss is separately computed for each router. For each router, given the frequency $t_i$ of how many tokens are routed to the $i$-th expert and the routing score $s_i$, the load balancing loss is computed via:

$$\mathcal{L}^{\text{balance}} = \frac{N}{|\mathcal{B}|} \sum_{i=1}^{N} \sum_{\text{token} \in \mathcal{B}} t_i \frac{\exp(s_i/\tau_0)}{\sum_{j=1}^{N} \exp(s_j/\tau_0)}, \tag{9}$$

where $N$ is the number of the experts, $\mathcal{B}$ is a batch of training examples, $|\mathcal{B}|$ is the number of tokens, and $\tau_0$ stands for a constant temperature. Different from the learnable $\tau$ in Equation (8), $\tau_0$ is kept fixed during training. The overall training objective is to minimize:

$$\mathcal{L} = \mathcal{L}_{\text{task}} + \alpha \mathcal{L}^{\text{balance}}, \tag{10}$$

where $\alpha$ is a coefficient for load balancing . The term $\mathcal{L}_{\text{task}}$ is determined by the specific task that Transformer learns. For example, we employ the masked language modeling loss (Devlin et al., 2019) for pre-training, and the sequence-to-sequence learning objective for neural machine translation.

## 3.3 Frozen Routing During Fine-tuning

We evaluate SMoE under the pre-training-then-fine-tuning paradigm in our work. During fine-tuning, we freeze all the parameters of experts, including both the router and expert networks. Because the fine-tuning datasets are usually small compared with pre-training corpora. We find that SMoE models tend to overfit downstream tasks, which often leads to inconsistent routing. Freezing SMoE parameters helps to relieve the above issues. Notice that we still use load balancing loss although the routers are kept fixed, which empirically improves fine-tuning performance in our experiments.

# 4 Experiments

We conduct experiments on cross-lingual language model pre-training (Devlin et al., 2019). We evaluate the performance by fine-tuning the pretrained models on various downstream benchmarks. We also compare validation losses of the masked language modeling task. Our method is named as X-MoE in the following sections.

## 4.1 Experimental Setup

**Pre-training Data**   Following (Chi et al., 2021), we use the combination of CCNet (Wenzek et al., 2019) and Wikipedia dump as pre-training corpora. We sample sentences in $94$ languages from the corpora, and employ a re-balanced distribution introduced by Conneau and Lample (2019), which increases the probability of low-resource languages.

**Model Architecture and Hyperparameters**   We construct our X-MoE models using the Transformer (Vaswani et al., 2017) encoder (L = 12, H = 768, A = 12) with the vocabulary provided by Conneau et al. (2020) as the backbone architecture. Following Lewis et al. (2021), we build a 32-expert sparse layer with 3 FFN sub-layers, and insert it after the 6-th Transformer layer. The routing dimension $d_e$ is set as 16. The gating temperature $\tau_0$ is set as $0.3$ and $0.07$ for the softmax gate and sigmoid gate, respectively. The detailed hyperparameters of X-MoE models can be found in Appendix A. X-MoE models are pretrained with the Adam optimizer ($\beta_1 = 0.9$, $\beta_2 = 0.98$) using a batch size of $2,048$ for 125K steps. The pre-training procedure takes 2 days on 2 Nvidia DGX-2 Stations. Appendix B and Appendix C provide the detailed hyperparameters for X-MoE pre-training and fine-tuning.

**Baselines**   We consider two baselines in our experiments. (1) **Dense** is a dense Transformer encoder without sparsely-activated modules. (2) **SMoE** is our implementation of Switch Transformers (Fedus et al., 2021). The SMoE baseline is built with the same setting with X-MoE. In addition to its original softmax-gating implementation, we also implement a sigmoid-gating (Lewis et al., 2021; Dai et al., 2022) variant of Switch Transformers as a baseline approach. Notice that the baseline models are pretrained with the same training data as X-MoE for a fair comparison.

Table 1: Evaluation results on the cross-lingual XTREME benchmark. The models are fine-tuned on the English training data and directly evaluated in all target languages. SMoE models are grouped according to the choice of gating function. The results are averaged over five runs.

| Model | Structured Prediction | | Question Answering | | | Classification | | Avg |
| | POS | NER | XQuAD | MLQA | TyDiQA | XNLI | PAWS-X | |
|---|---|---|---|---|---|---|---|---|
| Metrics | F1 | F1 | F1 / EM | F1 / EM | F1 / EM | Acc. | Acc. | |
| Dense (without SMoE) | 70.0 | 61.1 | 67.3 / 51.1 | 58.7 / 41.1 | 42.1 / 28.3 | 70.1 | 84.1 | 61.4 |
| *With* softmax *gating* | | | | | | | | |
| SMoE Baseline | 70.1 | 60.9 | 71.3 / 55.2 | 62.8 / 44.8 | 50.9 / 34.5 | 71.5 | 84.6 | 63.8 |
| X-MoE (Ours) | **70.8** | **63.2** | **72.4 / 56.2** | **64.5 / 46.3** | **53.7 / 38.1** | **72.0** | **85.2** | **65.3** |
| *With* sigmoid *gating* | | | | | | | | |
| SMoE Baseline | 70.6 | 61.2 | 71.5 / 55.7 | 63.2 / 45.3 | 50.1 / 35.1 | 71.2 | 85.1 | 64.1 |
| X-MoE (Ours) | **71.1** | **62.7** | **72.3 / 56.3** | **64.3 / 46.0** | **51.5 / 36.6** | **72.2** | **85.2** | **65.0** |

Table 2: Results of upstream evaluation. We report the validation perplexities on masked language modeling.

| Model | Perplexity |
|---|---|
| Dense (without SMoE) | 23.51 |
| *With* softmax *gating* | |
| SMoE Baseline | 19.02 |
| X-MoE (Ours) | **18.72** |
| *With* sigmoid *gating* | |
| SMoE Baseline | 19.59 |
| X-MoE (Ours) | **19.12** |

Table 3: Ablation studies of X-MoE components. The models employ various combinations of dimension reduction, $L_2$ normalization, and frozen routing. Average fine-tuning results of five random seeds are reported.

| Dim. Red. | $L_2$ Norm | Frozen | XNLI | MLQA |
|---|---|---|---|---|
| ✓ | ✓ | ✓ | **72.2** | **64.3 / 46.0** |
| ✗ | ✓ | ✓ | 71.5 | 63.4 / 45.2 |
| ✓ | ✗ | ✓ | 71.4 | 63.0 / 45.2 |
| ✓ | ✓ | ✗ | 71.7 | 63.9 / 45.8 |
| ✗ | ✗ | ✓ | 71.6 | 63.6 / 45.5 |
| ✗ | ✗ | ✗ | 71.2 | 63.2 / 45.3 |

## 4.2 Downstream Evaluation

We conduct a downstream evaluation on seven widely-used cross-lingual understanding benchmarks from XTREME (Hu et al., 2020). Specifically, we conduct experiments on Universal Dependencies v2.5 part-of-speech tagging (Zeman et al., 2019), WikiAnn named entity recognition (Pan et al., 2017; Rahimi et al., 2019), natural language inference (XNLI; Conneau et al. 2018), paraphrase adversaries from word scrambling (PAWS-X; Yang et al. 2019), and question answering on MLQA (Lewis et al., 2020), XQuAD (Artetxe et al., 2020), and TyDiQA-GoldP (Clark et al., 2020). Among the benchmarks, we adopt the cross-lingual transfer setting, where the models are fine-tuned with the training data in English and evaluated in all target languages.

Table 1 presents the evaluation results on the seven downstream tasks from the XTREME benchmark. For each task, the results are first averaged among the test languages and then averaged over five random seeds. Overall, the softmax-gating X-MoE model obtains the best performance, achieving an average score of 65.3. Comparing SMoE models with the dense model, SMoE models show notable improvement, indicating that SMoE models benefit from the large model capacity. Comparing X-MoE with the two SMoE baselines, it shows that X-MoE models provide consistent gains on downstream tasks, demonstrating the effectiveness of our proposed routing algorithm. We also validate X-MoE under the top-2 routing setting. Table 7 presents the evaluation results on XNLI, showing consistent improvements over the baseline for both top-1 and top-2 routing settings.

## 4.3 Upstream Evaluation

We compare the pretrained models for the upstream performance by the validation perplexity on masked language modeling (MLM). We sample multilingual sentences from mC4 (Xue et al., 2020), and construct an MLM validation dataset that contains $65,536$ sequences with lengths around $512$.

The results are shown in Table 2. Similar to the downstream results, we observe that SMoE models perform better than the dense model. In terms of the SMoE models, X-MoE models with both softmax

Table 4: BLEU scores on multilingual machine translation on WMT-10. The models are evaluated in the directions of 'x → en'.

| Model | ro | fr | cs | et | de | hi | tr | fi | lv | Avg |
|---|---|---|---|---|---|---|---|---|---|---|
| Dense (without SMoE) | 36.9 | 33.7 | 29.8 | 27.8 | 40.6 | 25.4 | 24.6 | 22.2 | 20.9 | 29.1 |
| SMoE Baseline | 37.1 | 33.8 | 31.0 | 28.6 | **42.3** | 26.0 | 24.3 | 23.0 | 21.2 | 29.7 |
| X-MoE | **37.9** | **34.3** | **31.1** | **29.0** | 42.2 | **27.0** | **24.8** | **23.5** | **21.6** | **30.2** |

Table 5: Comparison of routing dimensions for dimensionality reduction. '$N$' stands for the number of experts.

| Routing Dim. | XNLI | MLQA |
|---|---|---|
| $N/4$ | 71.4 | **64.3 / 46.4** |
| $N/2$ | **72.2** | **64.3** / 46.0 |
| $N$ | 71.7 | 63.8 / 45.9 |
| $2N$ | 71.7 | 62.7 / 44.8 |
| $4N$ | 71.2 | 63.1 / 45.0 |

Table 6: Effects of load balancing during fine-tuning. The models are fine-tuned with various weights for the auxiliary load balancing loss.

| Weight | XNLI | MLQA |
|---|---|---|
| 0 | 71.71 | 64.57 |
| $10^{-3}$ | 71.65 | 64.40 |
| $10^{-2}$ | **71.93** | **64.59** |
| $10^{-1}$ | 71.68 | 64.50 |

Table 7: Evaluation results on XNLI under the top-1 and top-2 routing settings. The models use the softmax gating functions.

| Model | top-$K$ | XNLI |
|---|---|---|
| SMoE Baseline | 1 | 71.5 |
| SMoE Baseline | 2 | 73.4 |
| X-MoE | 1 | 72.0 |
| X-MoE | 2 | **73.7** |

and sigmoid gating achieve lower masked language modeling perplexities than their counterparts. Among all the pretrained models, the softmax-gating X-MoE the achieves the lowest validation perplexity. The results show that our method not only works well for learning transferable text representations for downstream tasks, but also brings improvements to the upstream masked language modeling task. Comparing the upstream results with the downstream results, it shows that achieving a lower upstream perplexity does not promise better downstream performance. For instance, the sigmoid-gating X-MoE model has larger perplexity than the softmax-gating SMoE baseline has, but outperforms the fine-tuning performance of the baseline on the downstream tasks.

We also conduct experiments on the multilingual machine translation task. As shown in Table 4, we present the BLEU scores on the WMT-10 (Wang et al., 2020) dataset where the models are evaluated in the directions of 'x → en'. X-MoE consistently outperforms both the dense model and the SMoE baseline in eight translation directions.

## 4.4 Ablation Studies

**Routing Algorithm** To better understand our routing algorithm, we pretrain several variants of sigmoid-gating X-MoE models with various combinations of dimension reduction (Dim. Red.), $L_2$ normalization ($L_2$ Norm), and routing frozen (Frozen). For a fair comparison, all the models are pretrained and fine-tuned under the same setup, i.e., training data, steps, and the random seeds. We evaluate the models on XNLI and MLQA, and report the results in Table 3. Jointly using the three routing methods achieves the best performance. When ablating one of the three routing methods, the model performs less well, demonstrating that X-MoE benefits from all the three components.

**Dimension of Expert Embedding** We conduct experiments by adjusting the routing dimension for dimensionality reduction. Specifically, we compare sigmoid-gating X-MoE models with routing dimensions of $N/4$, $N/2$, $N$, $2N$, and $4N$, where $N$ is the number of the experts. Table 5 shows the downstream performance. It shows that using the routing dimension of $N/2$ provides the best performance for XNLI and $N/4$ is the best for MLQA. The results also confirm that dimension reduction better fits in with the low-rank nature of SMoE routing.

**Load Balancing During Fine-tuning** We explore whether load balancing is beneficial for fine-tuning SMoE models. To this end, we add load balancing loss to the total loss with various weights when fine-tuning X-MoE models on XNLI and MLQA. Table 6 shows the average validation scores where we search the load balancing coefficient $\alpha$ ranging from 0 to $10^{-1}$. We observe that using balance loss during fine-tuning is slightly beneficial for X-MoE. When removing the balance loss, X-MoE still remains comparable results on both XNLI and MLQA.

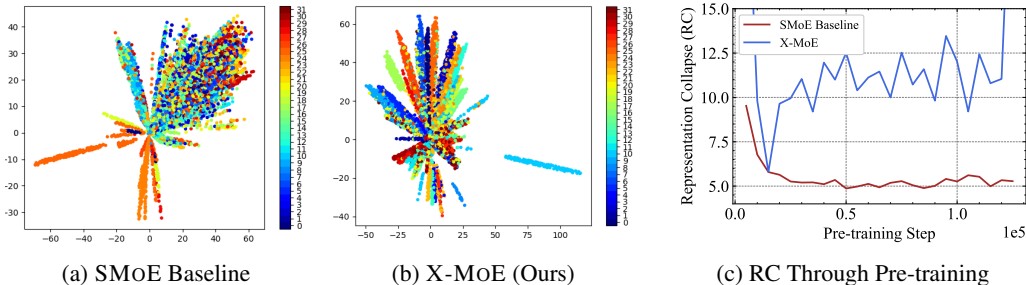

|  (a) SMoE Baseline | (b) X-MoE (Ours) | (c) RC Through Pre-training |

Figure 2: Analysis on the representation collapse of the Transformer hidden states. Figure (a) and (b) visualize the spatial structure of the experts. Each data point represents a token to be routed, and its color stands for the expert that it is assigned to. Figure (c) presents the curves of representation collapse (RC), which measures the within-class variability of hidden states. Larger RC values indicate less collapse.

## 4.5 Analysis

**Representation Collapse**   We qualitatively analyze the representation collapse issue by visualizing the experts. Figure 2a and 2b illustrate the spatial structure of the experts of SMoE baseline and X-MoE in hyperbolic space, which is produced by Uniform Manifold Approximation and Projection (UMAP; McInnes et al. 2018) with n-neighbor of 100 and min-dist of 1. Each data point represents a token to be routed, where we use the hidden states for SMoE baseline and the projected token representations for X-MoE. Each color stands for an expert that the tokens are assigned to.

Figure 2a shows that most of the points are mixed together with a large amount of available room unused, which suggests a representation collapse in the expert embedding space. In contrast, X-MoE in Figure 2b shows a well-organized feature space with clear distinctions between clusters. It indicates that our routing methods successfully project the tokens to the expert embedding space with routing features preserved.

Additionally, we conduct quantitative analysis on the degree of representation collapse for the learned Transformer hidden states that are fed into SMoE routing. We use the representation collapse metric proposed in (Zhu et al., 2021). Given the representations to be measured, we use $\Sigma_W$ and $\Sigma_B$ to denote the within-class and between-class covariance matrices, respectively. The representations collapse (RC) metric is calculated via:

$$\text{RC} = \text{Tr}(\Sigma_W \Sigma_B^\dagger), \tag{11}$$

where $\Sigma_B^\dagger$ is the pseudo inverse of $\Sigma_B$. Smaller RC values indicate representation collapse to a greater extent. Figure 2c illustrates the metrics during pre-training, where the data is sampled from the validation set mentioned in Section 4.3. SMoE baseline is unlike unconstrained feature models that can empirically collapse to almost zero RC, but still shows a consistent descending trend through pre-training, implying a trend toward representation collapse. Differently, X-MoE obtains larger RC scores than SMoE baseline with uptrend through pre-training.

**Routing Consistency Through Pre-training**   We examine whether our proposed routing algorithm achieves more consistent routing through training. We measure the routing consistency via the routing fluctuation (RF) ratio metric. Routing fluctuation is defined as the change of the target expert of an input token. Correspondingly, the RF ratio measures the ratio of RF between the current and the last checkpoints for the same input. A lower RF ratio indicates better routing consistency. As shown in Figure 3a, we present the RF ratio on the MLM validation set mentioned in Section 4.3. After the 15K step, X-MoE shows a much lower RF ratio than the SMoE baseline, indicating that our model produces more consistent routing behaviors.

**Inter-run Consistency Through Fine-tuning**   In the experiments of the downstream evaluation, we find that the routing behaviors of SMoE baseline models can be sensitive to random seeds. As the learned token assignments are various for different training data orders, the final downstream performance can be diverse among runs. Therefore, we study the routing behaviors of the SMoE

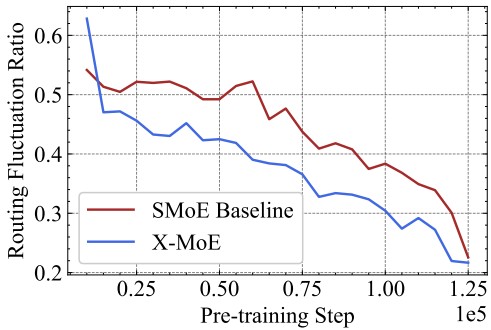

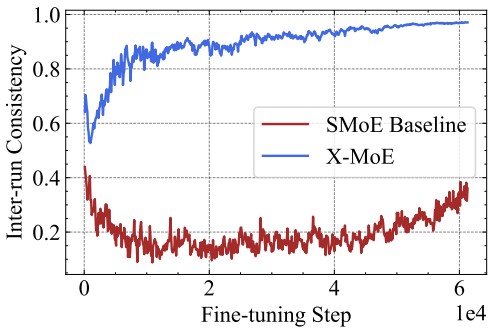

(a) Routing Fluctuation Ratio Through Pre-training     (b) Inter-run Consistency Through Fine-tuning

Figure 3: The routing behaviors of SMoE baseline and X-MoE. (a) Routing fluctuation (RF) ratio measures the ratio of the tokens that change their target experts between two checkpoints. Smaller RF values indicate more stable routing. (b) Inter-run consistency measures the correlation among the token assignments of various fine-tuning runs. Larger values indicate more consistent routing.

baseline and X-MoE models through fine-tuning. To achieve this, we develop a metric, named *inter-run consistency*, which measures how closely the token assignments converge among the runs with different seeds. Considering a model with $N$ experts, let $l = [n_1, ..., n_N]$ denote the total load of the experts, where $n_i$ stands for the number of the tokens that are assigned to the $i$-th expert. Given two loads $l_1$ and $l_2$ from two runs with different seeds, the similarity between $l_1$ and $l_2$ is defined as the Pearson correlation coefficient (PCC) between them, which is denoted as $\rho_{l_1, l_2}$. Here PCC only serves as a similarity metric rather than measuring linear correlation between variables. By extending it to $m$ runs with different seeds for each run, we define the inter-run consistency as the average of correlation matrix $\text{IC} = \sum_{i,j \in \{1...m\}} \rho_{l_i, l_j} / m^2$.

We fine-tune X-MoE and SMoE baseline models on XNLI for 12 runs separately. Then we compute the inter-run consistency for every 100 mini-batches, i.e., the expert loads are accumulated for 100 steps. Figure 3b illustrates the inter-run consistency. The SMoE baseline converges toward different routing solutions across multiple runs of fine-tuning, even though the only difference between runs is the random seed. In comparison, X-MoE obtains substantially better inter-run consistency than the SMoE baseline. The curve of X-MoE indicates that the models have various routing behaviors at the beginning of the fine-tuning, but finally converge to almost the same routing behaviors.

## 5   Related Work

**SMoE for Large-Scale Models**  Sparse Mixture-of-Experts (SMoE) models are introduced by Shazeer et al. (2017), which extends mixture of experts (Jacobs et al., 1991; Jordan and Jacobs, 1994) with conditional computation (Bengio et al., 2013; 2015) techniques. Taking advantage of computational computation, SMoE enables a massive increase in model capacity while maintaining computational efficiency. To explore the potential of SMoE, recent studies apply SMoE in a wide range of machine learning problems such as machine translation (Lepikhin et al., 2021), image classification (Riquelme et al., 2021), speech recognition (Kumatani et al., 2021). In addition to the supervised learning scenario, there has been work on exploring SMoE under the pre-training-fine-tuning paradigm, and observing discrepancies between strong pre-training quality and poor fine-tuning performance (Fedus et al., 2021; Artetxe et al., 2021; Zoph et al., 2022). Besides, the scaling behaviors of SMoE are also studied (Clark et al., 2022; Du et al., 2021).

**SMoE Routing Algorithms**  Many recent studies explore the token assignment algorithms for SMoE routing. BASE layers (Lewis et al., 2021) formulate the token routing problem as a linear assignment problem. Hash Layers (Roller et al., 2021) employ a parameter-free assignment algorithm that routes tokens by hashing. Zhou et al. (2022) let each expert select top-k tokens rather than distribute tokens to experts. Dai et al. (2022) propose to freeze the routing function in order to relieve routing fluctuation. These methods focus on the assignment algorithm in routing, but our routing algorithm focuses on improving the underlying routing scoring metric, which is still under-explored.

**Representation Collapse** Representation collapse, also termed neural collapse, is the degeneration of the representations during the training of neural networks. Several studies observe that the within-class variation of the representations in classification networks becomes negligible at the terminal phase of training (Papyan et al., 2020; Zhu et al., 2021; Tirer and Bruna, 2022). Besides, this phenomenon has also been observed in language model fine-tuning (Aghajanyan et al., 2021), and visual representation learning (Chen and He, 2021; Ermolov et al., 2021; Jing et al., 2022). These studies focus on densely-activated neural networks. In this work, we point out the representation collapse issue in SMoE models.

## 6  Conclusion

In this work, we point out the representation collapse issue in sparse mixture-of-experts (SMoE) models, and propose a routing algorithm that estimates the routing scores on a low-dimensional hypersphere. We conduct extensive experiments on cross-lingual language model pre-training. Experimental results across various benchmarks demonstrate that our method brings consistent improvements over SMoE baselines in terms of both language modeling and fine-tuning performance. Besides, our method alleviates the trend toward representation collapse and achieves more consistent routing. We are going to improve the work from the following perspectives. First, most current X-MoE experiments are conducted on language tasks, such as multilingual language model pre-training, and machine translation. We will also evaluate the proposed method on vision pretraining (Bao et al., 2022; Peng et al., 2022) and multimodal pretraining (Wang et al., 2022). Second, we would like to report the results of scaling up model size. The performance gain tends to be greater with a larger number of experts.

**Ethical Considerations** One of the negative societal impacts of training large-scale models is the high computational and environmental cost. Our paper focuses on improving SMoE, which is usually more efficient than dense model training with the same number of parameters. So better SMoE algorithms potentially save required computation and lessen $CO2$ emissions from computing. Moreover, X-MoE improves multilingual pre-training and fine-tuning, so that we can better transfer cross-lingual knowledge from high- to low-resource languages. The bless of larger model size brought by SMoE reduces the parameter conflicts of multilinguality, while keeping the computation cost manageable.

## Acknowledgement

We would like to acknowledge Bo Zheng and Zhiliang Peng for the helpful discussions. Heyan Huang is the corresponding author. Zewen Chi, Xian-Ling Mao, and Heyan Huang are supported by National Key R&D Plan (No. 2020AAA0106600), National Natural Science Foundation of China (No. U21B2009, 62172039, and L1924068).

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
