## A  Model Hyperparameters

Table 1 presents the model hyperparameters of X-MoE. The gating temperature $\tau_0$ is initialized as 0.3 and 0.07 for the softmax gating and sigmoid gating, respectively. We use the same vocabulary as XLM-R (Conneau et al., 2020) with 250K subwords tokenized by SentencePiece (Kudo and Richardson, 2018).

Table 1: Model hyperparameters of X-MoE.

| Hyperparameters | Value |
|---|---|
| FFNs in X-MoE layer | 3 |
| Number of experts | 32 |
| Expert embedding dimension | 16 |
| Initialized gating temperature $\tau_0$ | 0.3 / 0.07 |
| Transformer blocks | 12 |
| Hidden size | 768 |
| FFN inner hidden size | 3,072 |
| Attention heads | 12 |

## B  Hyperparameters for Pre-training

Table 2 presents the hyperparameters for pre-training.

Table 2: Hyperparameters for pre-training.

| Hyperparameters | Value |
|---|---|
| Optimizer | Adam |
| Training steps | 125,000 |
| Batch tokens per task | 1M |
| Adam $\epsilon$ | 1e-6 |
| Adam $\beta$ | (0.9, 0.98) |
| Maximum learning rate | 5e-4 |
| Learning rate schedule | Linear decay |
| Warmup steps | 10,000 |
| Weight decay | 0.01 |
| Transformer dropout | 0.1 |
| X-MoE dropout | 0 |
| Load balancing coefficient | 1e-2 |

## C  Hyperparameters for Fine-tuning

Table 3 presents the hyperparameters for fine-tuning.

Table 3: Hyperparameters for fine-tuning on the XTREME downstream tasks.

| Hyperparameters | POS | NER | XQuAD | MLQA | TyDiQA | XNLI | PAWS-X |
|---|---|---|---|---|---|---|---|
| Batch size | 8 | 8 | 32 | 32 | 32 | 32 | 32 |
| Learning rate | {1,2,3}e-5 | {5,...,9}e-6 | {2,3,4}e-5 | {2,3,4}e-5 | {2,3,4}e-5 | 2e-5 | {1,2}e-5 |
| LR schedule | Linear decay | Linear decay | Linear decay | Linear decay | Linear decay | Linear decay | Linear decay |
| Warmup | 10% | 10% | 10% | 10% | 10% | 20% | 10% |
| Weight decay | 0 | 0 | 0 | 0 | 0 | 1e-2 | 0 |
| Epochs | 10 | 10 | {2,3,4} | {2,3,4} | {10,20,40} | 5 | 10 |
| Load balancing | 0,1e{-3,-2,-1} | 0,1e{-3,-2,-1} | 0,1e{-3,-2,-1} | 0,1e{-3,-2,-1} | 0,1e{-3,-2,-1} | 0,1e{-3,-2,-1} | 0,1e{-3,-2,-1} |