# OpenReview forum: "On the Representation Collapse of Sparse Mixture of Experts"
_NeurIPS.cc/2022/Conference — NeurIPS 2022 Accept_

### Official Review · Reviewer_5h1o · 2022-07-11

**Rating:** 7
**Confidence:** 4
**Soundness:** 3 good
**Presentation:** 3 good
**Contribution:** 2 fair

**Summary:**

This paper looks at the problem of representational collapse in Mixture Of Experts (MoE) based transformer layers. Specifically, they show that for a softmax based gating function (for combining the MoE layer output with the hidden representation from the previous layer) , the gradient updates to token representations consists of a term that is simply a linear combination of the $N$ expert vectors, and since $N$ is much less than the dimensionality of the token vectors themselves, this portion of the gradient update is “low-dimensional” (specifically, $<= N$).

The paper argues that this could lead to representational collapse where token vectors receive updates that do not fully utilize the entire dimensionality of the vectors, leading to poorly learnt representations. In response, the paper proposes a simple (and somewhat effective) fix for this: Simply project the representations into a lower dimensional space before feeding it into the MoE layer. This ensures that the token vectors “interact” with the MoE layers (and hence with the corresponding gradients) via the low dimensional mapping. From experiments, they show that this simple fix results in representations with much lesser representational collapse (figure 2c) and in-turn modest gains on down-stream performance (Table-1)

**Questions:**

* Method: Could the authors clarify the role of other technical ``bells and whistles’’ for preventing representational collapse: using cosine similarity instead of dot-products (L2 normalization), using a temperature parameter in the softmax? I can see these being applied to the baselines as well - was this also done?
* Experiments: Could the authors comment on why the MoE parameters were kept frozen while finetuning? I only somewhat buy the overfitting argument because pre-trained transformers have been finetuned on extremely small data without much “overfitting” and several tricks can be employed to prevent it (such as early stopping, using low learning rates etc).

Please also see the weaknesses section from above for a more comprehensive list of questions on experiments.

**Limitations:**

Limitations adequately addressed.

**Strengths And Weaknesses:**

Strengths:
* Paper makes a clever observation regarding representational collapse in mixture of experts models, which to the best of my knowledge, has not been studied or pointed out previously.
* Conditional computation is becoming increasingly important for scaling to ever larger model sizes and thus the overall theme of the paper is extremely relevant.
* The experiments on cross-lingual NLP show some modest imrprovements.

Weaknesses:
* Method: This work only shows representational collapse to be a problem for softmax gating, but this doesn’t seem to be the case for sigmoid? In particular, the contribution from the $J_2$ term only ends up pushing the token embedding towards $e_k$ (the chosen expert). So while one can say that all tokens that are assigned expert $k$ will receive a gradient which has a direction along $e_k$ (hence some amount of clustering), the $J_1$ term will ensure that they also get a gradient pointing in a different direction. (This second comment also applies to the case when you use a softmax gating).
* Experiments:
    * Maybe I misunderstood the model, but it seems like only a single MoE layer is added after layer 6, instead of having several MoE layers (perhaps every other layer is an MoE layer). The reason why i ask this is because if there are more MoE layers, that increases the chances of representational collapse, and could probably show bigger gains for your approach. If there is only one MoE layer, then the representational collapse might not be that beneficial.
    * On the flipside, it would be great to have a toy experiment on simple toy data (even linear regression) where you have only a single MoE layer with a softmax gating, where you can clearly demonstrate representational collapse.
    * It’s unclear whether the difference b/w Dense (no MoE) and MoE models is because of the MoE layers or just because of more parameters i.e. did you ensure that the Dense models have the same number of parameters?
    * The UMAP plots don’t seem to suggest representational collapse: It’s really hard to conclude that 2a and 2b have different amounts of spread in the representations. It appears that for both of these, tokens that are mapped to the same expert are very close together. The plot in fig2c is nice but I would like really like to see more analysis / quantitative results on representational collapse. Some suggestions:
        * you could sample a large enough number of tokens and see if the subspace spanned by them is larger for your approach vs the baseline (presumably, for the baseline you’d expect the subspace to be much lower dimensional).
        * you could also look at spread i.e. given a cluster of all token vectors that are assigned the same expert, what is thethe average L2 distance of each token vector from the mean?

---

> ### Author Response · Authors · 2022-08-02
> **Response Part 1**
>
> We thank the reviewer for the valuable comments. We are happy to see that the reviewer agreed with the representation collapse problem and confirm the importance of our work.
>
> **Question1**
> > "This work only shows representational collapse to be a problem for softmax gating, but this doesn’t seem to be the case for sigmoid?"
>
> In the case of sigmoid, the term $\boldsymbol  J_2$ is not related to the other experts except the chosen one. Then, Equation (5) can be simplified as
> $\boldsymbol  J_2^\top \nabla_{\boldsymbol  h'} \mathcal{L} = S_{k}(1 - S_{k})(\boldsymbol  h^{{FFN}\top}\nabla_{\boldsymbol  h'} \mathcal{L})  \boldsymbol  e_{k},$
> which is a special case of the original Equation (5) when $j \in \\{k\\}$.
>
> **Question2**
> > "Maybe I misunderstood the model, but it seems like only a single MoE layer is added after layer 6, instead of having several MoE layers (perhaps every other layer is an MoE layer). The reason why i ask this is because if there are more MoE layers, that increases the chances of representational collapse, and could probably show bigger gains for your approach. If there is only one MoE layer, then the representational collapse might not be that beneficial."
>
> We follow the setting of BASE Layers[1] about how to insert MoE layers. The improvements are consistent across different insertion methods.
>
> **Question3**
> >  "It’s unclear whether the difference b/w Dense (no MoE) and MoE models is because of the MoE layers or just because of more parameters i.e. did you ensure that the Dense models have the same number of parameters?"
>
> We have included additional machine translation experiments where the models have the same number of activated parameters and require almost the same training FLOPs.  In Table R.2, we compare X-MoE with the dense Transformer and Switch Transformer. All the models have almost the same FLOPs. X-MoE outperforms both baseline models on multilingual machine translation.
>
> **Table R.2**: BLEU scores on WMT-10 multilingual machine translation in directions of x->en. '# act params' means the number of the activated parameters per token.
> | Models | #act params | # params | FLOPs | avg | de | ro | fr | cs | et | hi | tr | fi | lv |
> | ----- | ---- | ---- | ---- | ---- | ---- | ---- | ---- | ---- | ---- | ---- | ---- | ---- | ---- |
> | Dense Transformer | 90M | 90M | 317B | 29.1 | 40.6 | 36.9 | 33.7 | 29.8 | 27.8 | 25.4 | 24.6 | 22.2 | 20.9 |
> | Switch Transformer | 90M | 480M | 317B | 29.7 | **42.3** | 37.1 | 33.8 | 31 | 28.6 | 26 | 24.3 | 23 | 21.2 |
> | X-MoE | 90M | 480M | 317B | **30.16** | 42.2 | **37.9** | **34.3** | **31.1** | **29** | **27** | **24.8** | **23.5** | **21.6** |
>
> **Question4**
> > "The UMAP plots don’t seem to suggest representational collapse: It’s really hard to conclude that 2a and 2b have different amounts of spread in the representations. It appears that for both of these, tokens that are mapped to the same expert are very close together. The plot in fig2c is nice but I would like really like to see more analysis / quantitative results on representational collapse." Some suggestions:
> >-   you could sample a large enough number of tokens and see if the subspace spanned by them is larger for your approach vs the baseline (presumably, for the baseline you’d expect the subspace to be much lower dimensional).
> > -   you could also look at spread i.e. given a cluster of all token vectors that are assigned the same expert, what is thethe average L2 distance of each token vector from the mean?
>
> We thank the reviewer for the constructive suggestions. Following the suggestions, we look at the spread by computing the average L2 distance of each token hidden vector from the mean for each expert, where we use the validation data mentioned in Section 4.3. Then, we compute the weighted average over experts. Besides, we also compute the average token-vector covariance trace for each expert. The results are shown in Table R.3. X-MoE has larger L2 and variance scores, indicating that X-MoE alleviates the trend toward representation collapse.
>
> **Table R.3**
>  | Models | L2 | Variance |
>  | ----- | ----- | ----- |
>  | SMoE Baseline |  9.19 | 89.08 |
>  | X-MoE | **9.69** | **96.21** |
>
> **Questions5**
> >  "Method: Could the authors clarify the role of other technical 'bells and whistles' for preventing representational collapse: using cosine similarity instead of dot-products (L2 normalization), using a temperature parameter in the softmax? I can see these being applied to the baselines as well - was this also done?"
>
> We tried the components in baseline in Table 3 as ablation studies.
>
> **Questions6**
> > "Experiments: Could the authors comment on why the MoE parameters were kept frozen while finetuning?"
>
> We conducted ablation studies in Table 3, where freezing MoE parameters improves the downstream performance.
>
>
> **References**
>
> [1] BASE Layers: Simplifying Training of Large, Sparse Models

---

> > ### Comment · Reviewer_5h1o · 2022-08-07
> > **Thanks for the response!**
> >
> > Thank you for promptly adding in these experiments! This clarifies many points for me, and I also think the "spread" experiment should be moved into the main section.

---

### Official Review · Reviewer_gm6K · 2022-07-11

**Rating:** 7
**Confidence:** 4
**Soundness:** 3 good
**Presentation:** 3 good
**Contribution:** 3 good

**Summary:**

This paper makes the following contributions:
1. Identify that common MoE routing mechanisms that rely on matching token representations to expert representations suffer from representation collapse - learnt token representations become too similar to expert representations collapsing to the subspace of expert representations.
2. Propose a simple fix to the representation collapse problem - utilizing a low rank projection and cosine similarity metric for routing.
3. Evaluate their approach and baseline routing algorithms on multilingual text MLM pre-training and downstream evaluation on the Xtreme benchmark.
4. Qualitative and quantitative analysis demonstrating that the approach presented in the paper improves the representation collapse in MoEs.

**Questions:**

Please see weaknesses above.

**Strengths And Weaknesses:**

Strengths:
1. Identify a previously unknown representation collapse problem in Sparse Mixture-of-Experts models. Propose and evaluate a simple modification to the routing architecture that addresses the problem.
2. Careful experimentation evaluating the approach on multilingual text MLM pre-training and on downstream tasks from the Xtreme benchmark.
3. Intuitive qualitative and quantitative analysis of the learnt token representations that highlight the collapse and that the proposed modification addresses the problem.
4. The paper is well motivated, clearly written and makes it easy to understand the details of the experiments.

Weaknesses:
1. Limited downstream evaluations: The downstream evaluations on the Xtreme benchmark only evaluate on English-only fine-tuning and zero-shot evaluations. The zero-shot evaluation setting is a little brittle and can benefit from regularization effects and minor fine-tuning differences. It would strengthen the results to: (i) include language-level evaluation results (perhaps in the appendix). (ii) Conduct evaluations in non zero-shot settings perhaps using translated data to simulate multilingual fine-tuning (Translate-Train).
2. Non-standard upstream evaluation: The upstream evals evaluate MLM perplexity on a custom validation split. This is not a standard upstream evaluation setting and would be hard to replicate if there are minor differences in experimental setups or vocabularies or test split. I would advise the authors to repeat these experiments on a more standard benchmark; perhaps a causal LM with perplexity evaluations on standard test sets.
3. Limited experimental setting: The paper analyzes and evaluates MoEs trained with Top-1 routing. While theoretically you would expect generalized MoEs trained with Top-K routing to suffer from similar representational collapse, it would strengthen the paper to validate that their approach helps when MoEs are trained with Top-2 routing as well.


Overall I recommend weak accept. This paper identifies and addresses a problem with the sparse-MoE architecture which could be valuable to large model training for a variety of different tasks and applications. However, there are several limitations with their evaluation and the scope of experiments presented in the paper.

EDIT: Updated recommendation to accept following author rebuttal. The proposed algorithm seems to result in small but consistent improvements for MoE training on multilingual text pre-training and Machine Translation.

---

> ### Author Response · Authors · 2022-08-02
> **Response Part 1**
>
> We are delighted to see reviewers confirm the representation collapse problem and agree the work is well motivated. We appreciate the constructive suggestions on experiments.
>
> **Question1**
> > "Limited downstream evaluations: The zero-shot evaluation setting is a little brittle and can benefit from regularization effects and minor fine-tuning differences. It would strengthen the results to ... "
>
> We have conducted additional experiments to strengthen the results.
>
> **Top-2 routing**
> We have added new experiments for the top-2 routing setting.  We use Gshard[2] as the baseline. We pretrain both Gshard and X-MoE models with top-2 routing using the replaced token detection task for 50K steps.  Then, we fine-tune both models on the XNLI task. As shown in Table R.1, X-MoE outperforms Gshard on XNLI, which justifies that our methods also work for the top-2 routing setting.
>
> **Table R.1**: Top-2 routing results on XNLI.
> | Models | avg | en | fr | es | de | el | bg | ru | tr | ar | vi | th | zh | hi | sw | ur |
> | ----- | ----- | ----- | ----- | ----- | ----- | ----- | ----- | ----- | ----- | ----- | ----- | ----- | ----- | ----- | ----- | ----- |
> | Gshard (top-2)| 73.38 | 85.25 | 78.45 | 79.35 | 76.55 | 74.05 | 75.9 | 74.85 | 73.3 | 70.2 | 75.65 | 71.2 | 74 | **68.15** | 59.55 | 64.2 |
> | X-MoE (top-2) | **73.72** | **85.5** | **78.57** | **79.43** | **76.7** | **74.13** | **76.67** | **75.23** | **73.53** | **70.9** | **75.67** | **72.17** | **74.43** | 68.03 | **60.33** | **64.57** |
>
> **Machine translation**
> We have added experiments on machine translation. In Table R.2, we compare X-MoE with the dense Transformer and Switch Transformer. All the models have almost the same FLOPs. X-MoE outperforms both the baseline models on multilingual machine translation.
>
> **Table R.2**: BLEU scores on WMT-10 multilingual machine translation in directions of x->en. '# act params' means the number of the activated parameters per token.
> | Models | #act params | # params | FLOPs | avg | de | ro | fr | cs | et | hi | tr | fi | lv |
> | ----- | ---- | ---- | ---- | ---- | ---- | ---- | ---- | ---- | ---- | ---- | ---- | ---- | ---- |
> | Dense Transformer | 90M | 90M | 317B | 29.1 | 40.6 | 36.9 | 33.7 | 29.8 | 27.8 | 25.4 | 24.6 | 22.2 | 20.9 |
> | Switch Transformer | 90M | 480M | 317B | 29.7 | **42.3** | 37.1 | 33.8 | 31 | 28.6 | 26 | 24.3 | 23 | 21.2 |
> | X-MoE | 90M | 480M | 317B | **30.16** | 42.2 | **37.9** | **34.3** | **31.1** | **29** | **27** | **24.8** | **23.5** | **21.6** |
>
> **Language-level evaluation results**
>
> Following the suggestions of including language-level evaluation results, we will include the language-level evaluation results as follows (see Table R.4 - R.10).
>
> **Question2**
> > "Non-standard upstream evaluation: The upstream evals evaluate MLM perplexity on a custom validation split. This is not a standard upstream evaluation setting and would be hard to replicate if there are minor differences in experimental setups or vocabularies or test split. I would advise the authors to repeat these experiments on a more standard benchmark; perhaps a causal LM with perplexity evaluations on standard test sets."
>
> The upstream evaluation is for apple-to-apple comparisons between the methods of the paper. For easier replication, we have also included the machine translation results (Table R.2) that are based on public data.

---

> > ### Author Response · Authors · 2022-08-02
> > **Part 2**
> >
> > **Question3**
> >
> > > "Limited experimental setting: The paper analyzes and evaluates MoEs trained with Top-1 routing. While theoretically you would expect generalized MoEs trained with Top-K routing to suffer from similar representational collapse, it would strengthen the paper to validate that their approach helps when MoEs are trained with Top-2 routing as well."
> >
> >
> > 1. Our methods are applicable to not only top-1 routing but also top-k routing[1,2].
> > 2. We have added experiments, which show that our method helps for top-2 routing as well.
> >
> > - Theoretically, the representation collapse issue we study is a general problem in SMoE models no matter whether they are top-1 or top-k routing models. When using top-k routing, the forward function of SMoE is $k_1, k_2 \dots k_K = {topk}_i(s_i) = {topk}_i(\boldsymbol {h}\cdot\boldsymbol {e}_i)$,
> >
> >    $f^{SMoE}(\boldsymbol  h) = \boldsymbol {h} + \sum_{i=1 \dots K} g(s_{k_i})f_{k_i}^{FFN}(\boldsymbol  h)$,
> >
> >    where the gating function is defined as $g(s_{k_i})={exp}(s_{k_i}) / \sum_{i=1 \dots K} {exp}(s_{k_i})$, according to [1,2]. As the expert outputs are linearly combined, Equation (5) holds for the top-k routing, which can be written as $\boldsymbol  J_2^\top \nabla_{\boldsymbol  h'} \mathcal{L} = \sum_{i=1\dots K}c_i\boldsymbol  e_{k_i}$,
> >
> >     $c_i= \sum_{j=1\dots K} S_{k_j}(\delta_{{k_j}{k_i}} - S_{k_j})  (\boldsymbol  h^{{FFN}_j^T} \\nabla {\boldsymbol  h'} \mathcal{L})$.
> >
> >     Thus, the conclusion on representation collapse holds for the top-k routing.
> >
> >
> > - Methodologically, our methods can be directly applied to top-k routing.
> >
> > - We have added new experiments for the top-2 routing setting.  The results are shown in Table R.1
> >
> > **References**
> >
> > [1] Outrageously Large Neural Networks: The Sparsely-Gated Mixture-of-Experts Layer
> >
> > [2] GShard: Scaling Giant Models with Conditional Computation and Automatic Sharding

---

> > > ### Author Response · Authors · 2022-08-02
> > > **Language-level evaluation results (Part 1)**
> > >
> > > Following the suggestions of including language-level evaluation results, we will include the language-level evaluation reuslts as follows (see Table R.4 - R.10)
> > >
> > > **Table R.4** Language-level results on XNLI.
> > > | Models | Gating | avg | en | fr | es | de | el | bg | ru | tr | ar | vi | th | zh | hi | sw | ur |
> > > | ----- | ----- | ----- | ----- | ----- | ----- | ----- | ----- | ----- | ----- | ----- | ----- | ----- | ----- | ----- | ----- | ----- | ----- |
> > > Dense | N/A | 70.1 | 83.7 | 76.3 | 76.2 | 73.6 | 71.1 | 72.6 | 71.8 | 68.7 | 67.6 | 72.7 | 67.1 | 70.7 | 64.7 | 52.9 | 61.5
> > > SMoE Baseline | sigmoid | 71.5 | 84.3 | 76.5 | 77.7 | 75.4 | 71.9 | 74.0 | 73.1 | 70.2 | 68.9 | 73.6 | 69.2 | 71.7 | 65.6 | 58.1 | 62.4
> > > X-MoE | softmax | 72.0 | 84.7 | 77.1 | 78.0 | 75.0 | 72.6 | 74.4 | 73.8 | 70.9 | 69.7 | 74.2 | 70.2 | 72.1 | 66.5 | 58.1 | 63.4
> > > SMoE Baseline | sigmoid | 71.2 | 83.5 | 76.6 | 77.0 | 74.2 | 72.3 | 73.9 | 72.8 | 69.6 | 68.8 | 73.0 | 68.7 | 71.5 | 65.5 | 57.8 | 62.3
> > > X-MoE | sigmoid | 72.2 | 84.2 | 77.6 | 78.3 | 75.4 | 73.2 | 74.7 | 74.2 | 71.0 | 69.8 | 74.0 | 70.4 | 72.7 | 66.7 | 56.3 | 63.7
> > >
> > >
> > > **Table R.5** Language-level results on PAWS-X.
> > > | Models | Gating | avg | en | fr | de | es | ja | ko | zh |
> > > ---- | ---- | ---- | ---- | ---- | ---- | ---- | ---- | ---- | ---- |
> > > Dense | N/A | 84.1 | 94.4 | 87.6 | 86.1 | 87.6 | 76.7 | 75.7 | 80.6
> > > SMoE Baseline | sigmoid | 84.6 | 94.7 | 88.9 | 87.4 | 88.6 | 77.4 | 75.8 | 79.4
> > > X-MoE | softmax | 85.2 | 94.6 | 89.4 | 88.4 | 89.1 | 77.3 | 76.5 | 81.1
> > > SMoE Baseline | sigmoid | 85.1 | 94.7 | 89.5 | 88.0 | 88.3 | 77.2 | 76.4 | 81.3
> > > X-MoE | sigmoid | 85.2 | 94.9 | 89.0 | 88.1 | 88.1 | 78.2 | 77.5 | 80.6
> > >
> > >
> > > **Table R.6** Language-level results on XQuAD.
> > > | Models | Gating | avg | en | es | de | el | ru | tr | ar | vi | th | zh | hi |
> > > ---- | ---- | ---- | ---- | ---- | ---- | ---- | ---- | ---- | ---- | ---- | ---- | --- | ---- |
> > > Dense | N/A | 67.3/51.1 | 83.8/71.6 | 71.8/56.4 | 72.3/58.9 | 68.2/49.3 | 70.9/52.9 | 57.6/41.4 | 59.1/41.9 | 69.5/50.0 | 61.2/48.9 | 69.2/50.9 | 56.2/39.9 |
> > > SMoE Baseline | softmax | 71.3/55.2 | 84.8/73.5 | 76.1/61.3 | 75.4/61.8 | 71.8/53.0 | 73.6/56.9 | 64.3/48.5 | 67.0/50.1 | 73.4/53.4 | 64.3/50.4 | 70.7/50.8 | 62.9/47.2 |
> > > X-MoE | softmax | 72.4/56.2 | 84.8/73.3 | 77.9/63.8 | 75.1/62.1 | 70.8/51.0 | 74.7/57.8 | 66.2/50.4 | 67.4/50.4 | 73.5/53.8 | 65.6/52.1 | 74.0/54.3 | 66.0/49.4 |
> > > SMoE Baseline | sigmoid | 71.5/55.7 | 83.9/72.2 | 77.5/62.4 | 73.5/59.9 | 70.9/52.4 | 73.7/57.2 | 65.5/49.9 | 65.0/47.8 | 73.5/53.1 | 65.7/54.9 | 73.4/53.1 | 64.1/49.3 |
> > > X-MoE | sigmoid | 72.3/56.3 | 84.5/72.7 | 77.4/62.3 | 75.7/62.2 | 68.8/48.6 | 74.7/58.0 | 65.9/49.9 | 66.8/49.2 | 74.4/54.7 | 67.8/56.3 | 74.8/56.6 | 64.5/48.4 |
> > >
> > > **Table R.7** Language-level results on MLQA.
> > > | Models | Gating | avg | en | es | de | ar | hi | vi | zh |
> > > ---- | ---- | ---- | ---- | ---- | ---- | ---- | ---- | ---- | ---- |
> > > Dense | N/A | 58.7/41.1 | 80.0/66.6 | 61.7/43.6 | 58.3/43.4 | 48.6/30.1 | 48.0/32.1 | 60.2/40.3 | 54.5/31.8 |
> > > SMoE Baseline | sigmoid | 62.8/44.8 | 80.7/67.4 | 66.0/47.2 | 60.8/45.8 | 54.8/35.5 | 54.0/37.4 | 64.6/44.6 | 58.6/35.4 |
> > > X-MoE | softmax | 64.5/46.3 | 81.1/67.8 | 67.4/49.2 | 62.0/46.7 | 56.3/36.6 | 57.5/40.6 | 65.4/45.0 | 62.1/38.6 |
> > > SMoE Baseline | sigmoid | 63.2/45.3 | 80.5/67.3 | 66.3/48.3 | 61.5/46.1 | 54.0/34.8 | 54.6/38.6 | 64.6/43.9 | 60.7/37.9 |
> > > X-MoE | sigmoid | 64.3/46.0 | 80.6/67.3 | 67.0/48.8 | 62.2/46.8 | 54.8/35.3 | 57.2/40.4 | 66.0/45.2 | 62.1/38.6 |
> > >
> > > **Table R.8** Language-level results on TyDiQA.
> > > | Models | Gating | avg | en | ar | bn | fi | id | ko | ru | sw | te |
> > > ---- | ---- | ---- | ---- | ---- | ---- | ---- | ---- | ---- | ---- | ---- | ---- |
> > > Dense | N/A | 42.1/28.3 | 64.9/51.4 | 48.8/29.1 | 30.4/19.6 | 47.5/29.9 | 53.7/38.5 | 30.9/22.4 | 46.5/27.4 | 36.0/22.5 | 20.5/13.8 |
> > > SMoE Baseline | sigmoid | 50.9/34.5 | 66.0/53.8 | 55.8/36.1 | 40.8/25.5 | 54.7/37.4 | 61.8/45.8 | 40.8/28.3 | 57.0/33.1 | 45.9/27.2 | 35.4/23.3 |
> > > X-MoE | softmax | 53.7/38.1 | 67.9/55.0 | 59.1/38.5 | 44.0/31.7 | 59.8/43.6 | 65.7/49.8 | 47.9/35.4 | 57.1/33.8 | 49.2/32.5 | 32.6/22.9 |
> > > SMoE Baseline | sigmoid | 50.1/35.1 | 66.0/52.6 | 54.9/35.2 | 33.8/20.9 | 53.7/36.4 | 60.4/45.0 | 45.2/35.1 | 55.4/34.7 | 47.2/30.1 | 33.8/25.6 |
> > > X-MoE | sigmoid | 51.5/36.6 | 67.1/54.2 | 57.2/37.2 | 35.3/23.9 | 58.8/43.1 | 62.3/45.7 | 43.6/32.2 | 57.0/37.2 | 48.7/30.7 | 33.9/25.1 |

---

> > > > ### Author Response · Authors · 2022-08-02
> > > > **Language-level evaluation results (Part 2)**
> > > >
> > > > **Table R.9** Language-level results on POS.
> > > > | Models | Gating | avg | af | ar | bg | de | el | en | es | et | eu | fa | fi | fr | he | hi | hu | id | it | ja | kk | ko | mr | nl | pt | ru | ta | te | th | tl | tr | ur | vi | yo | zh |
> > > > | ---- | ---- | ---- | ---- | ---- | ---- | ---- | ---- | ---- | ---- | ---- | ---- | ---- | ---- | ---- | ---- | ---- | ---- | ---- | ---- | ---- | ---- | ---- | ---- | ---- | ---- | ---- | ---- | ---- | ---- | ---- | ---- | ---- | ---- | ---- | ---- |
> > > > Dense | N/A | 70.0 | 86.3 | 65.6 | 86.4 | 87.0 | 84.7 | 95.4 | 87.9 | 79.8 | 62.9 | 68.6 | 79.1 | 86.6 | 64.6 | 67.5 | 80.3 | 71.7 | 88.6 | 17.8 | 70.9 | 53.3 | 69.6 | 88.6 | 86.9 | 88.6 | 61.4 | 75.0 | 41.9 | 79.3 | 72.3 | 57.9 | 55.2 | 23.8 | 25.4 |
> > > > SMoE Baseline | softmax | 70.1 | 87.0 | 65.3 | 87.5 | 87.5 | 86.3 | 95.6 | 87.3 | 81.1 | 65.3 | 67.2 | 80.8 | 86.2 | 64.1 | 64.0 | 79.7 | 72.0 | 88.2 | 14.2 | 73.1 | 52.4 | 70.2 | 89.4 | 86.9 | 88.2 | 62.3 | 76.6 | 39.1 | 80.9 | 73.6 | 55.9 | 57.0 | 24.3 | 23.7 |
> > > > X-MoE | softmax | 70.8 | 87.8 | 63.9 | 87.6 | 88.0 | 84.5 | 95.6 | 87.8 | 81.5 | 64.4 | 68.1 | 81.1 | 86.2 | 63.5 | 67.8 | 80.4 | 71.7 | 88.7 | 18.1 | 73.1 | 53.0 | 73.4 | 89.5 | 87.5 | 88.4 | 61.7 | 77.2 | 40.2 | 84.6 | 73.7 | 59.5 | 57.1 | 23.1 | 27.8 |
> > > > SMoE Baseline | sigmoid | 70.6 | 86.9 | 66.1 | 87.8 | 88.1 | 85.8 | 95.5 | 88.3 | 81.1 | 64.0 | 68.3 | 80.9 | 87.0 | 64.5 | 64.5 | 80.7 | 72.0 | 88.9 | 17.7 | 72.5 | 52.4 | 70.0 | 89.4 | 87.5 | 88.4 | 61.4 | 75.9 | 39.6 | 86.5 | 73.4 | 55.7 | 56.7 | 24.2 | 29.3 |
> > > > X-MoE | sigmoid | 71.1 | 87.7 | 66.6 | 88.4 | 88.0 | 85.2 | 95.5 | 88.6 | 81.7 | 64.0 | 68.6 | 81.1 | 87.6 | 65.8 | 68.3 | 81.4 | 72.3 | 89.1 | 16.7 | 73.0 | 52.7 | 70.8 | 89.3 | 88.0 | 89.1 | 61.7 | 75.4 | 38.1 | 88.0 | 73.7 | 59.7 | 57.0 | 25.4 | 26.4 |
> > > >
> > > >
> > > >
> > > > **Table R.10** Language-level results on NER.
> > > > Models | Gating | avg | ar | he | vi | id | jv | ms | tl | eu | ml | ta | te | af | nl | en | de | el | bn | hi | mr | ur | fa | fr | it | pt | es | bg | ru | ja | ka | ko | th | sw | yo | my | zh | kk | tr | et | fi | hu |
> > > > | ---- | ---- | ---- | ---- | ---- | ---- | ---- | ---- | ---- | ---- | ---- | ---- | ---- | ---- | ---- | ---- | ---- | ---- | ---- | ---- | ---- | ---- | ---- | ---- | ---- | ---- | ---- | ---- | ---- | ---- | ---- | ---- | ---- | ---- | ---- | ---- | ---- | ---- | ---- | ---- | ---- | ---- | ---- |
> > > > Dense | N/A | 61.1 | 43.6 | 52.2 | 71.9 | 55.7 | 50.5 | 52.0 | 73.4 | 54.7 | 60.2 | 57.7 | 52.0 | 76.7 | 80.9 | 82.9 | 77.4 | 71.7 | 73.5 | 69.7 | 61.1 | 57.8 | 40.6 | 78.7 | 79.0 | 78.9 | 75.8 | 76.5 | 62.7 | 19.1 | 68.1 | 55.8 | 4.4 | 67.9 | 38.3 | 49.8 | 24.9 | 52.4 | 76.2 | 71.4 | 75.7 | 73.7 |
> > > > SMoE Baseline | softmax | 60.9 | 42.1 | 51.1 | 69.9 | 58.9 | 54.6 | 66.1 | 72.5 | 57.6 | 58.2 | 54.5 | 52.6 | 77.2 | 80.9 | 83.0 | 76.7 | 74.4 | 72.6 | 65.2 | 60.0 | 45.5 | 36.6 | 79.0 | 79.0 | 80.0 | 73.9 | 77.3 | 65.1 | 12.6 | 66.9 | 55.7 | 2.1 | 67.8 | 38.2 | 51.8 | 17.0 | 55.9 | 77.0 | 73.2 | 76.3 | 75.7 |
> > > > X-MoE | softmax | 63.2 | 48.5 | 55.2 | 70.6 | 62.3 | 50.6 | 67.6 | 73.3 | 53.7 | 57.8 | 59.5 | 58.1 | 78.2 | 82.7 | 83.3 | 78.4 | 76.9 | 74.5 | 68.2 | 62.4 | 52.7 | 38.3 | 78.8 | 81.4 | 79.2 | 76.5 | 78.6 | 67.1 | 16.5 | 72.9 | 59.9 | 2.9 | 67.9 | 48.7 | 60.2 | 22.9 | 50.6 | 78.8 | 75.2 | 77.8 | 79.5 |
> > > > SMoE Baseline | sigmoid | 61.2 | 43.4 | 53.0 | 71.1 | 54.8 | 52.0 | 61.8 | 74.0 | 54.4 | 61.7 | 56.9 | 51.9 | 78.0 | 81.3 | 83.2 | 76.9 | 74.3 | 72.7 | 68.0 | 62.9 | 59.1 | 37.2 | 77.0 | 79.9 | 77.8 | 66.7 | 77.4 | 62.6 | 12.1 | 68.2 | 58.9 | 2.4 | 69.5 | 41.6 | 53.7 | 15.4 | 55.4 | 75.1 | 73.0 | 76.1 | 75.5 |
> > > > X-MoE | sigmoid | 62.7 | 45.5 | 55.1 | 72.7 | 63.6 | 54.2 | 55.4 | 73.9 | 60.9 | 61.1 | 55.8 | 56.3 | 79.6 | 82.6 | 83.4 | 77.4 | 78.3 | 74.8 | 68.1 | 62.9 | 48.5 | 36.7 | 79.8 | 80.1 | 77.6 | 76.6 | 77.8 | 67.7 | 16.2 | 69.1 | 58.6 | 2.8 | 66.6 | 47.5 | 53.6 | 19.4 | 57.9 | 78.9 | 75.8 | 77.2 | 78.5 |

---

> > > > > ### Comment · Reviewer_gm6K · 2022-08-06
> > > > > **Thanks for the response**
> > > > >
> > > > > Thanks for addressing my comments. I have updated my recommendation to accept.

---

### Official Review · Reviewer_JmFj · 2022-07-18

**Rating:** 3
**Confidence:** 5
**Soundness:** 1 poor
**Presentation:** 2 fair
**Contribution:** 2 fair

**Summary:**

Paper proposes a remedy to the representation collapse issue of sparsely gated (activated) mixture of expert modes (sMoE) in Transformers. The hypothesis, and later supported with observations is that, the top-1 routing mechanism pushes the hidden representations clustering around expert centroids, introducing a tendency toward representation collapse - which might be hurting the model accuracy.

To mitigate the representation collapse, deemed to be harmful, paper proposes adding two more operations to the routing procedure: (1) dimensionality reduction of the hidden representations that the routing decisions used to base on, and (2) L2 normalization of the activations (logits) of pre-softmax/sigmoid at the top-1 expert selection routine.

Two enhancements combined, paper demonstrates improved upstream (pretraining perplexity) and downstream (accuracy after fine-tuning)  quality gains over the dense and regular top-1 sMoE models on various benchmarks.

**Questions:**

Q1 - What does "conservative" mean in line 128?

Q2 - What are the initial and final values of \tau (in line 131) that are learned by the model - does the model learn higher or lower values than the one initially set?

Q3 - In line 142, you mention "we employ ... seq2seq for nmt" but there is no nmt experiments in the paper, is this intended?

Q4 - What is the alpha / temperature for the language mixing (section 4.1 - Pre-training Data)?

Q5 - You mention in 4.3 (last sentence) that the sigmoid gating has different behavior in upstream evaluation, but do not provide any data points / tables / figures - have I missed those?

Q6 - Representation collapse (and others in Figure 2 & 3) are using which set to collect data? Dev / test / train ? Please specify.

Q7 - Why do you think routing consistency should compare the first and one of the last checkpoints to measure the stats? What is the hypothesis there? The model may change the chosen of expert throughout the course of training, which should be okay (please correct me if I'm wrong), rather than committing to an expert at the beginning of the training and sticking with it. Wouldn't it make more sense to take a look at the last n-checkpoints and computing the statistics over that? Please word your hypothesis why comparing first with the last checkpoint supports the argument for routing consistency.

Q8 - All models seem to have trained for the same number of tokens, but some of them use extra parameters and extra priors (dim-reductions + L2 norm). Given these, wouldn't it make more sense to compare two models with the same upstream (pretraining) perplexity on their downstream quality? How do we make sure the #tokens that we set in the pretraining not biasing the proposed method over the others ?

Q9 - Follow up on Q8, have we adjusted the learning rate schedules for all the different approaches (a la chinchilla) ?


**Limitations:**

adequate

**Strengths And Weaknesses:**

Strengths:

- Identification of "representation collapse" of MoE models as one of the potential issue of these family of models sheds light on the failure modes of them.

- New metrics, like routing fluctuation ratio, and UMAP of token-expert assignment, are novel and great contributions to the community moving forward.

Weakness:

- Proposed method is focusing only on top-1 expert choice sMoE models which are not widely used, and might be attempting to solve a problem that only exist in that space. Which is okay, but the paper is not scoped, structured and narrated that way. Do we know, the widely used top-2 expert choice, where the first expert is chosen according to eq. (3) and the second is sampled randomly from the same distribution, has the same problem?

- The assumption that the number of experts are going to be much smaller than the hidden dimension does not always hold (e.g., Lepikhin et al. 2020 use 2048 experts w/ 1024 hidden dimensions). Similar to the first weakness, scoping of the problem is one of the weaknesses.

---

> ### Author Response · Authors · 2022-08-02
> **Response Part 1**
>
> We thank the reviewer for the insightful comments.
>
> **About top-1/top-2 routing**
>
> > "Proposed method is focusing only on top-1 expert choice sMoE models which are not widely used, and might be attempting to solve a problem that only exist in that space. Which is okay, but the paper is not scoped, structured and narrated that way. Do we know, the widely used top-2 expert choice, where the first expert is chosen according to eq. (3) and the second is sampled randomly from the same distribution, has the same problem?"
>
> 1. Our methods are applicable to not only top-1 routing but also top-k routing[1,2].
> 2. Top-1 routing is widely-used in SMoE models, including language models (e.g., Switch Transformers[3], Hash Layers[4], BASE Layers[5]), visual models (e.g., Tutle[6]), multimodal models (e.g., LIMoE[7]), etc.
>
> - Theoretically, the representation collapse issue we study is a general problem in SMoE models no matter whether they are top-1 or top-k routing models. When using top-k routing, the forward function of SMoE is $k_1, k_2 \dots k_K = {topk}_i(s_i) = {topk}_i(\boldsymbol {h}\cdot\boldsymbol {e}_i)$,
>
>    $f^{SMoE}(\boldsymbol  h) = \boldsymbol {h} + \sum_{i=1 \dots K} g(s_{k_i})f_{k_i}^{FFN}(\boldsymbol  h)$,
>
>    where the gating function is defined as $g(s_{k_i})={exp}(s_{k_i}) / \sum_{i=1 \dots K} {exp}(s_{k_i})$, according to [1,2]. As the expert outputs are linearly combined, Equation (5) holds for the top-k routing, which can be written as $\boldsymbol  J_2^\top \nabla_{\boldsymbol  h'} \mathcal{L} = \sum_{i=1\dots K}c_i\boldsymbol  e_{k_i}$,
>
>     $c_i= \sum_{j=1\dots K} S_{k_j}(\delta_{{k_j}{k_i}} - S_{k_j})  (\boldsymbol  h^{{FFN}_j^T} \\nabla {\boldsymbol  h'} \mathcal{L})$.
>
>     Thus, the conclusion on representation collapse holds for the top-k routing.
>
> - Methodologically, our methods can be directly applied to top-k routing.
>
> - We have added new experiments for the top-2 routing setting[2].  We use Gshard[2] as the baseline. We pretrain both Gshard and X-MoE models with the replaced token detection task for 50K steps.  Then, we fine-tune both models on the XNLI task. As shown in Table R.1, X-MoE outperforms Gshard on XNLI, which justifies that our methods also work for the top-2 routing setting.
>
> **Table R.1**: Top-2 routing results on XNLI.
> | Models | avg | en | fr | es | de | el | bg | ru | tr | ar | vi | th | zh | hi | sw | ur |
> | ----- | ----- | ----- | ----- | ----- | ----- | ----- | ----- | ----- | ----- | ----- | ----- | ----- | ----- | ----- | ----- | ----- |
> | Gshard (top-2)| 73.38 | 85.25 | 78.45 | 79.35 | 76.55 | 74.05 | 75.9 | 74.85 | 73.3 | 70.2 | 75.65 | 71.2 | 74 | **68.15** | 59.55 | 64.2 |
> | X-MoE (top-2) | **73.72** | **85.5** | **78.57** | **79.43** | **76.7** | **74.13** | **76.67** | **75.23** | **73.53** | **70.9** | **75.67** | **72.17** | **74.43** | 68.03 | **60.33** | **64.57** |
>
>
> **About the number of experts $N \ll d$ in practice**
>
> > "The assumption that the number of experts are going to be much smaller than the hidden dimension does not always hold (e.g., Lepikhin et al. 2020 use 2048 experts w/ 1024 hidden dimensions). Similar to the first weakness, scoping of the problem is one of the weaknesses."
>
> First, the above analysis shows that, for the top-k (k>1) routing, the token representation tends to update towards a linear combination of $k$ expert vectors. It implies a trend toward representation collapse from $\mathbb{R}^d$ to $\mathbb{R}^k$, where $k \ll d$ in practice. The 2048-expert example (Lepikhin et al. [2]) uses top-2 routing. Therefore, it is **not** out of the scope of the representation collapse. The experimental results in Table R.1 show that X-MoE outperforms Gshard (Lepikhin et al. [2]) for the top-2 routing setting.
>
> Second, recent work shows that expert data parallelism, which divides GPU cards into groups and works as data parallelism with replicated expert weights, can learn the SMoE models more efficiently. For example, V-MoE [8] reports that "we found the total number of experts E = 32 to be the sweet spot in our setting". In GLaM[9], it mentions that "the largest GLaM 64B/64Experts model was trained on 1,024 Cloud TPU-V4 chips". Therefore, $N \ll d$ is quite common in practice.

---

> > ### Author Response · Authors · 2022-08-02
> > **Part 2**
> >
> > **Question1**
> > > "Q1 - What does "conservative" mean in line 128?"
> >
> > The word "conservative" means that the gate value has a narrow range. For example, in sigmoid gating, if the routing scores are in [-1, 1], then the gate value will be in [0.27, 0.73], which is "conservative". The introduction of the temperature makes the range be like [-1, 1] so that the gates can be fully "open" or "closed".
> >
> > **Question2**
> > > "Q2 - What are the initial and final values of \tau (in line 131) that are learned by the model - does the model learn higher or lower values than the one initially set?"
> >
> > As mentioned in L165 and L531, the gating temperature $\tau_0$ is set as 0.3 and 0.07 for the two kinds of gates, respectively. The learnable $\tau$ is initialized with $\tau_0$. In the experiments, we found that our models learn lower $\tau$ than they are initialized (e.g., 0.3 -> 0.1 for sigmoid gating).
> >
> > **Question3**
> > > "Q3 - In line 142, you mention "we employ ... seq2seq for nmt" but there is no nmt experiments in the paper, is this intended?"
> >
> > We have added experiments on machine translation. In Table R.2, we compare X-MoE with the dense Transformer and Switch Transformer. All the models have almost the same FLOPs. X-MoE outperforms both the baseline models on multilingual machine translation.
> >
> > **Table R.2**: BLEU scores on WMT-10 multilingual machine translation in directions of x->en. '# act params' means the number of the activated parameters per token.
> > | Models | #act params | # params | FLOPs | avg | de | ro | fr | cs | et | hi | tr | fi | lv |
> > | ----- | ---- | ---- | ---- | ---- | ---- | ---- | ---- | ---- | ---- | ---- | ---- | ---- | ---- |
> > | Dense Transformer | 90M | 90M | 317B | 29.1 | 40.6 | 36.9 | 33.7 | 29.8 | 27.8 | 25.4 | 24.6 | 22.2 | 20.9 |
> > | Switch Transformer | 90M | 480M | 317B | 29.7 | **42.3** | 37.1 | 33.8 | 31 | 28.6 | 26 | 24.3 | 23 | 21.2 |
> > | X-MoE | 90M | 480M | 317B | **30.16** | 42.2 | **37.9** | **34.3** | **31.1** | **29** | **27** | **24.8** | **23.5** | **21.6** |
> >
> > **Question4**
> > > "Q4 - What is the alpha / temperature for the language mixing (section 4.1 - Pre-training Data)?"
> >
> > We use $\alpha=0.7$ for language mixing following previous work.
> >
> > **Question5**
> > > "Q5 - You mention in 4.3 (last sentence) that the sigmoid gating has different behavior in upstream evaluation, but do not provide any data points / tables / figures - have I missed those?"
> >
> > The last sentence means achieving lower upstream perplexity does not promise better downstream performance, rather than the sigmoid gating has different behaviors.
> >
> > **Question6**
> > > "Q6 - Representation collapse (and others in Figure 2 & 3) are using which set to collect data? Dev / test / train ? Please specify."
> >
> > In  Figure 2 and Figure 3a, we use the dev set for MLM, mentioned in L245 and L254. In Figure 3b, we directly measure the inter-run consistency during fine-tuning, i.e., the data are the XNLI training data.
> >
> > **Question7**
> > > "Q7 - Why do you think routing consistency should compare the first and one of the last checkpoints to measure the stats? What is the hypothesis there? The model may change the chosen of expert throughout the course of training, which should be okay (please correct me if I'm wrong), rather than committing to an expert at the beginning of the training and sticking with it. Wouldn't it make more sense to take a look at the last n-checkpoints and computing the statistics over that? Please word your hypothesis why comparing first with the last checkpoint supports the argument for routing consistency."
> >
> > Routing consistency is measured between two consecutive checkpoints (i.e., looking at the last two checkpoints). It does not compare the first and one of the last checkpoints.

---

> > > ### Author Response · Authors · 2022-08-02
> > > **Part 3**
> > >
> > > **Question8**
> > > > "Q8 - All models seem to have trained for the same number of tokens, but some of them use extra parameters and extra priors (dim-reductions + L2 norm). Given these, wouldn't it make more sense to compare two models with the same upstream (pretraining) perplexity on their downstream quality? How do we make sure the #tokens that we set in the pretraining not biasing the proposed method over the others ?"
> > >
> > > In contrast to the question, our models use fewer parameters than the baseline SMoE model because of dimension reduction. Specifically, the baseline SMoE requires $32\times 768=24,576$ parameters to represent experts, but our model requires $32 \times 16$ parameters for expert embeddings and $768 \times 16$ for dimension reduction, i.e., our model only uses $12,800$ ($52\\%$ of the baseline) parameters for representing experts. Moreover, the #tokens used in pretraining is not biased to any method, where all methods are fairly compared with the same setting used by the dense baseline. Our method consistently outperforms baseline SMoE when comparing different intermediate steps (i.e., different #tokens). The improvements are also consistent on both machine translation and top-2 routing as shown in Table R.1 and Table R.2.
> > >
> > > **Question9**
> > > > "Q9 - Follow up on Q8, have we adjusted the learning rate schedules for all the different approaches (a la chinchilla) ?"
> > >
> > > The learning rate was tuned for the baseline dense model. Then we follow the same learning rate schedules for fair comparisons.
> > >
> > >
> > > **References**
> > >
> > > [1] Outrageously Large Neural Networks: The Sparsely-Gated Mixture-of-Experts Layer
> > >
> > > [2] GShard: Scaling Giant Models with Conditional Computation and Automatic Sharding
> > >
> > > [3] Switch Transformers: Scaling to Trillion Parameter Models with Simple and Efficient Sparsity
> > >
> > > [4] Hash Layers For Large Sparse Models
> > >
> > > [5] BASE Layers: Simplifying Training of Large, Sparse Models
> > >
> > > [6] Tutle: Adaptive Mixture-of-Experts at Scale
> > >
> > > [7] Multimodal Contrastive Learning with LIMoE: the Language-Image Mixture of Experts
> > >
> > > [8] Scaling Vision with Sparse Mixture of Experts
> > >
> > > [9] GLaM: Efficient Scaling of Language Models with Mixture-of-Experts

---

> ### Author Response · Authors · 2022-08-05
> **Looking forward to your feedback**
>
> Dear Reviewer,
>
> We appreciate your constructive comments for helping us improve our paper in many aspects. We have provided detailed responses to address your concerns. This is a gentle reminder since the author-reviewer discussion's deadline (Aug 9) is approaching.
>
> So far we have not heard any feedback after posting our replies to your questions. We are eager to hear back from you about whether our replies and the newly added results have addressed some of your concerns. We hope to make full use of the remaining rebuttal time.
>
> Best,
>
> Authors

---

> ### Author Response · Authors · 2022-08-07
> **Response to Reviewer JmFj**
>
> Dear Reviewer JmFj,
>
> We appreciate your constructive comments for helping us improve our paper in many aspects. We have provided detailed responses to address your concerns. This is a gentle reminder since the author-reviewer discussion's deadline (Aug 9) is approaching.
>
> So far we have not heard any feedback after posting our replies to your questions. We are eager to hear back from you about whether our replies and the newly added results have addressed some of your concerns. We hope to make full use of the remaining rebuttal time.
>
> Best,
>
> Authors

---

### Public Comment · ~zhiyuan_zeng2 · 2023-04-27
**Question about the dimension reduction and representation collapse**

Hello, I have some questions about the methods of this paper. I understand the potential problem of the representation collapse in MoE, but I think it may be better to compare the value of $J_1$ and $J_2$ in gradient analysis to show the significance of the problem. Another question is about dimension reduction, the paper seems to forget to explain why dimension reduction can address the representation collapse. In my view, if we reduce the dimension of representation, the assignment scores of different tokens will be more correlated, which may cause unbalanced routing. For example, if the dimension is reduced to 1, the assignment scores will be strongly correlated.

---

> ### Public Comment · ~Viet_Dung_Nguyen2 · 2024-05-02
> **Response**
>
> I also have read the paper for a while and I agree with you on the need for importance assessment between $J_1$ and $J_2$. For the question on dimension reduction, I believe this can be explained if you calculate the Jacobian of the loss w.r.t $h$.

---

### Meta-Review · Area_Chair_8rVh · 2022-08-25

**Recommendation:** Accept
**Confidence:** Certain

**Metareview:**

This paper highlights that contemporary sparse Mixture-of-Experts (sMoE) models suffer from representation collapse in the gating mechanism. The paper then proposes a simple fix reducing the dimensionality of the gating representations and using cosine similarity. The paper shows qualitatively that the new representation suffers less from collapse and that model trained with the new algorithm exhibit small but consistent improvements when evaluated on cross-lingual language understanding and machine translation.

All of the reviewers had a concern that the method is tested only in the top-1 setting vs. the most common practice of using top-2 or more experts. The paper added experiments on top of GShard that alleviates this concern.

Another concern was limited scope of evaluation. The authors added MT results as well as in-language results, which showed the same trend of small, but consistent improvements.

As a result of the discussion, two of the three reviewers were happy of how their concerns were addressed and increased their score to recommend that the paper is accepted. The third reviewer has not been active in discussion.

**Award:**

No

---

### Decision · Program_Chairs · 2022-09-14

Accept